# Reaching the last mile with ivermectin mass drug administration against onchocerciasis: The case of Kwanware-Ottou persistent transmission focus in the Wenchi health district of Ghana

Rogers Nditanchou[1,2*], Akinola Stephen Oluwole[3], Sapana Basnet[4], Alexandre Chailloux[4], Judith Saare[5], David Agyemang[6], Sandra Adelaide King[6], Mike Yaw Osei-Atweneboana[7], Richard Selby[4], Joseph Opare[5], Louise Hamill[4], Joseph Nelson Siewe Fodjo[2], Veronique Verhoeven[2], Elena Schmidt[4], Robert Colebunders[2]

1 Sightsavers Cameroon Country Office, Bastos, Yaoundé, Cameroon, 2 Global Health Institute, University of Antwerp, Campus drie Eiken, Gouverneur Kinsbergen Centrum, Antwerpen, Antwerp, Belgium, 3 Nigeria Abuja Country Office, Maitama, Abuja, Nigeria, 4 Sightsavers, Haywards Heath, United Kingdom, 5 Neglected Tropical Diseases Programme, Ghana Health Service, PMB, Ministries, Accra, Ghana, 6 Sightsavers Ghana Country Office, Accra, Ghana, 7 The Council for Scientific and Industrial Research (CSIR), Accra, Ghana

* rnditanchou@sightsavers.org, nditanchou@yahoo.com

## Abstract

### Background

Despite over two decades of Community-Directed Treatment with Ivermectin (CDTI), onchocerciasis transmission persists in localized pockets in Ghana, particularly in the Kwanware-Ottou community within the Wenchi Health District. This study trialled a scalable approach to identifying context-specific barriers and solutions for improving CDTI effectiveness.

### Methodology/principal findings

A mixed-methods approach was employed, including Geographical Information System mapping, community consultation, census and treatment coverage evaluation, and qualitative assessments. These informed the participatory development of an Action Plan, which was implemented and evaluated across three sub-districts. Key challenges identified and addressed included poor data quality, high population mobility, remote settlements with accessibility issues, limited awareness, and inadequate number and deployment of community drug distributors. As a result, therapeutic coverage increased from 70.8% to 88.2. Seven out of eight communities with pre-intervention coverage below the recommended 65% threshold not only achieved but exceeded this target. Ultimately, all communities met the coverage goal. The intervention also improved data accuracy and quality, community engagement, and

**Data availability statement:** All relevant data are within the manuscript and its Supporting Information files.

**Funding:** This work received financial support from the Coalition for Operational Research on Neglected Tropical Diseases (COR-NTD), which is funded at The Task Force for Global Health primarily by the Bill & Melinda Gates Foundation (OPP1190754) and by the United States Agency for International Development through its Neglected Tropical Diseases Program. Under the grant conditions of the Foundation, a Creative Commons Attribution 4.0 Generic License has already been assigned to the Author Accepted Manuscript version that might arise from this submission (Grant No 274G to RN). The funders had no role in study design, data collection and analysis, decision to publish, or preparation of the manuscript.

**Competing interests:** The authors have declared that no competing interests exist.

adherence to directly observed treatment, while addressing systemic gaps in CDTI delivery.

## Conclusions/significance

This study demonstrates that a coordinated, locally adapted stimulus package can significantly enhance CDTI performance in areas of persistent onchocerciasis transmission. The approach presents a scalable model for similar endemic settings and aligns with the World Health Organization's 2021–2030 Roadmap for the elimination of Neglected Tropical Diseases.

## Author summary

Onchocerciasis, or river blindness, is a parasitic disease transmitted by blackflies. It remains a public health challenge in parts of Ghana despite decades of community treatment with ivermectin. In Kwanware-Ottou area within the Wenchi Health District, transmission has persisted for over 27 years. This study investigated why onchocerciasis continues to spread in this region and tested new strategies to improve treatment coverage. Using satellite imagery, community mapping and interviews, we identified hard-to-reach settlements and population groups often missed during treatment. A locally tailored action plan was developed and implemented with strong community involvement. Results showed a significant increase in treatment coverage and improved community participation. This approach demonstrates how combining local knowledge, technology, and participatory planning can help overcome persistent disease transmission. These findings offer a practical model for other regions facing similar challenges in eliminating onchocerciasis and other neglected tropical diseases.

## Background

Onchocerciasis, also commonly known as river blindness, is a neglected tropical disease (NTD), endemic in Ghana and many other sub-Saharan African countries. The disease is caused by a parasitic worm called *Onchocerca volvulus* transmitted to human through the bite of infected blackflies. It is extremely debilitating, leading to significant visual impairment and blindness, as well as intense itching and skin changes [1]. The World Health Organization (WHO) has set a goal of eliminating onchocerciasis transmission by 2030 in 12% of endemic countries [2]. This is to be achieved mainly through Mass Drug Administration (MDA) using Community-Directed Treatment with Ivermectin (CDTI) approach, with a minimum coverage of 65% of total or 80% of treatment eligible population for at least 12–15 annual rounds of treatment [3–5]. Ghana has aligned and endeavoured to achieve this goal, but persistent transmission in some areas is hindering progress.

Onchocerciasis was first identified in Ghana in 1875 [6,7]. Control efforts began in 1974 under the Onchocerciasis Control Programme (OCP), which implemented aerial larvicide spraying that continued until 2002 [8]. Mass treatment with ivermectin (IVM) began in 1992, initially delivered by teams of mobile health workers [9]. In 1997, the strategy shifted to CDTI, where communities took charge of drug distribution [9]. In 2009, a policy of biannual CDTI was introduced in hyper and meso-endemic areas [9]. Currently in Ghana, around 8.6 million people are at risk of onchocerciasis infection and need MDA (ESPEN, https://espen.afro.who.int/countries/ghana).

In Kwanware-Ottou community in Wenchi Health District (HD), located in the Tano-Ankobra Onchocerciasis Operational Transmission Zone (OTZ) [9] of Ghana, transmission persists despite more than 27 years of MDA [10,11]. Impact evaluations revealed onchocerciasis microfilarial (mf) prevalence of 5% in 2012 and 29% in 2017 among adults aged 20 and above. Additionally, a seroprevalence of 38% was found in children aged 5–10 years in the 2017 evaluation, proving recent infection. In a follow-up study in 2021, we found an mf prevalence of 36.6% (95% CI: 19.9% - 56.0%) in Ottou and 29.2% (95% CI: 14.6% - 49.8%) in Kwanware, with corresponding blackfly infectivity rates of 6.7 per thousand (‰) and 5.9‰ [11]. This confirms persistent transmission of onchocerciasis in Kwanware-Ottou community.

A qualitative assessment conducted during the 2021 investigation revealed the following factors contributing to the persistent transmission: NTD programme's inability to reach some at-risk populations, including those living in highly remote settlements and the seasonal mobility of some groups, including (illegal) miners. Additionally, issues of incomplete census data and individual inconsistency/non-adherence to treatment over the years were contributing as well [11].

In the 2021 investigation, we identified that the focus of persistent infection extended within a 10 km radius of Kwanware-Ottou [11]. This conclusion was based on the reduction of entomological, parasitological, and serological indices to undetectable levels at the 10 km radius distance from Ottou [11]. This is consistent with other studies and WHO guidance that suggest a 10–12 km radius as the smallest epidemiological unit for onchocerciasis transmission [4,12,13]. Targeting interventions within this defined area is expected to yield significant and sustained impact. Strengthening CDTI is a critical first step, given the demonstrated effectiveness of MDA [4,5]. In the absence of drug resistance, CDTI remains central to elimination strategies regardless of the underlying drivers of persistent transmission.

This study developed and piloted strategies to enhance CDTI within the Kwanware-Ottou transmission focus, aiming not only to address localized challenges but also to demonstrate an adaptable model for other persistent transmission foci in Ghana and similar endemic regions. We additionally present key challenges and recommendations for programmes seeking to implement these strategies elsewhere.

## Methods

### Ethics statement

Ethical clearances for this study were obtained from the Ghana Health Service Ethics Review Committee (Clearance No GHS-ERC: 006/09/23). Additionally, administrative authorization was granted by the Director General of Ghana Health Services. Community consultation, mobilization, and raising awareness were conducted, during which verbal consent was obtained from community leaders. All participants underwent an informed consent process, and written informed consent forms were signed before data collection. For individuals younger than 18 years, assent forms were signed by their parents or guardians. The study adhered to ethical standards in handling participant data and ensuring voluntary participation.

### Settings

The 10 km radius area of the Kwanware-Ottou transmission focus covers parts of Subsinso, Nsawkaw, and Boase sub-districts in the Wenchi, Tain, and Banda HDs of Ghana, respectively. This area is drained by a dense river network, with two main rivers, Subin and Tain, both having actual and potential blackfly breeding sites [10,11]. The main

occupations of people living in these districts are cash crop farming, particularly cashew nuts and fruits. Additionally, arti-sanal mining is a major activity in the Nsawkaw sub-district, while cattle rearing is primarily practiced in the Subinso and Boase areas in Wenchi and Banda HDs, respectively. The intervention was implemented at the sub-district level to align with health system smallest operational unit. These sub-districts have well-defined geographic boundaries for programme implementation and include all communities within or near the 10 km radius around Ottou community, ensuring compre-hensive coverage of the focus. See Fig 1.

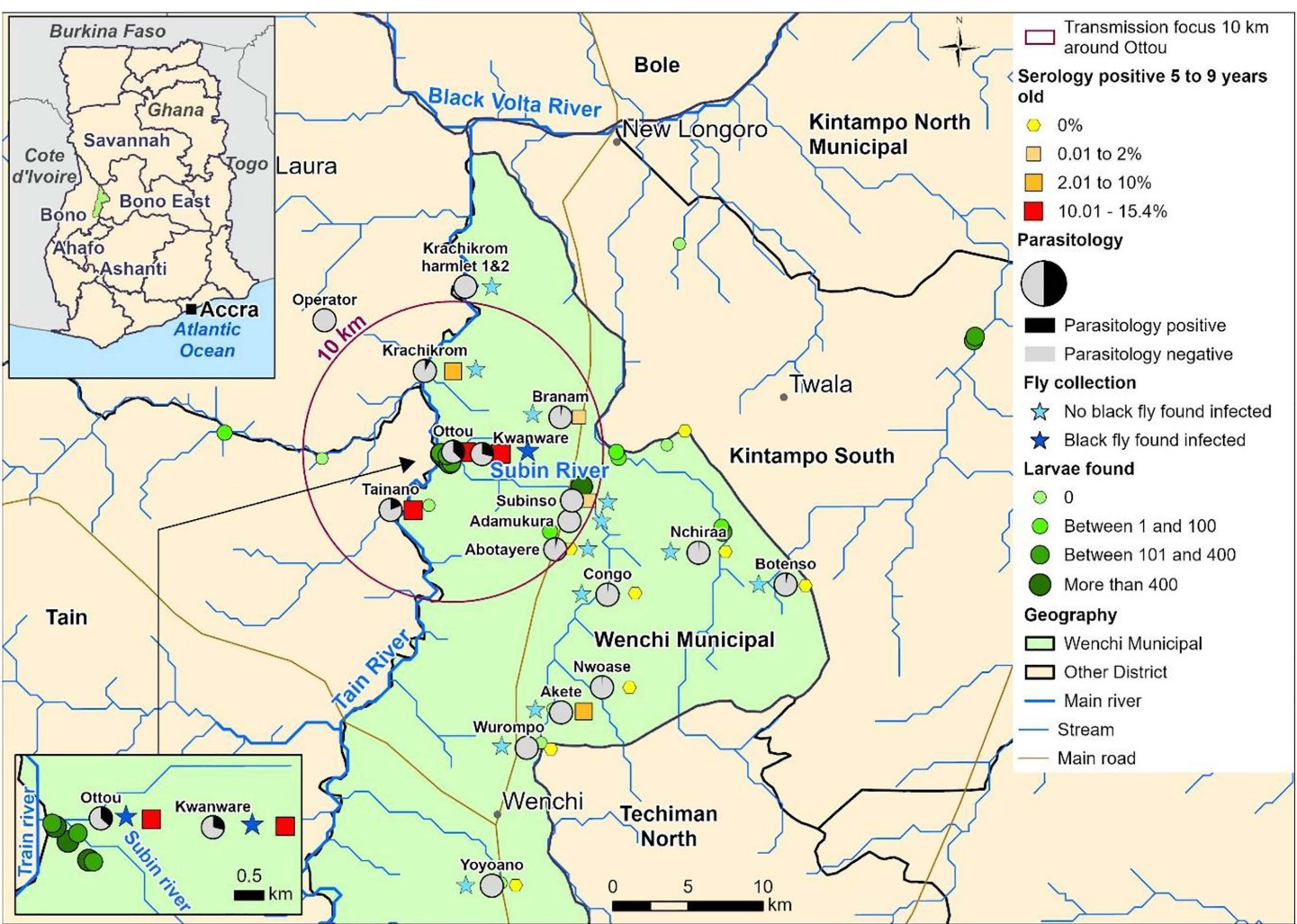

**Fig 1. Map of the study area showing a 10 km radius onchocerciasis transmission focus area around Ottou community.** Parasitology findings, shown as pie charts, indicate the proportion of individuals positive for microfilariae (mf+) detected via skin snip (March 2021). Serology findings are rep-resented by colour-graded squares, with size and shade reflecting Ov16 antibody prevalence (March 2021). Star symbols mark blackfly collection sites, with dark blue stars indicating detection of infective L3 larvae (October 2020). Coloured circles show locations where blackfly larvae were found. The size is proportional to the number of larvae detected. This map has been modified from our previous publication [11]. The base map uses national and district administrative boundaries from the geoBoundaries Global Database (https://www.geoboundaries.org/countryDownloads.html; license: CC BY 4.0, https://www.geoboundaries.org/metadata.html). Additional data sources include HydroATLAS hydrography (https://www.hydrosheds.org/hydroatlas; CC BY 4.0), manually digitized roads from high-resolution satellite imagery, and primary field-collected population and survey data. The map was created in ArcGIS Pro software using only openly licensed datasets.

## Study design

A mixed-method approach (quantitative and qualitative) was conducted from January to November 2024 to explore the implementation of CDTI in the study area, identifying challenges and solutions. This approach combined community consultations, Geographical Information System (GIS) technology, census including treatment coverage evaluation surveys, and qualitative assessments to elucidate context-specific challenges and solutions. These findings were subsequently used to guide the participatory development of an Action Plan (AP) aimed at improving CDTI effectiveness. The AP was then implemented by the programme. Fig 2 outlines the study design and procedure and Table 1 shows activity timeline.

**Procedure.** The strategies were implemented in six key steps, outlined below:

**Step 1: Identification of settlements through community consultations.**

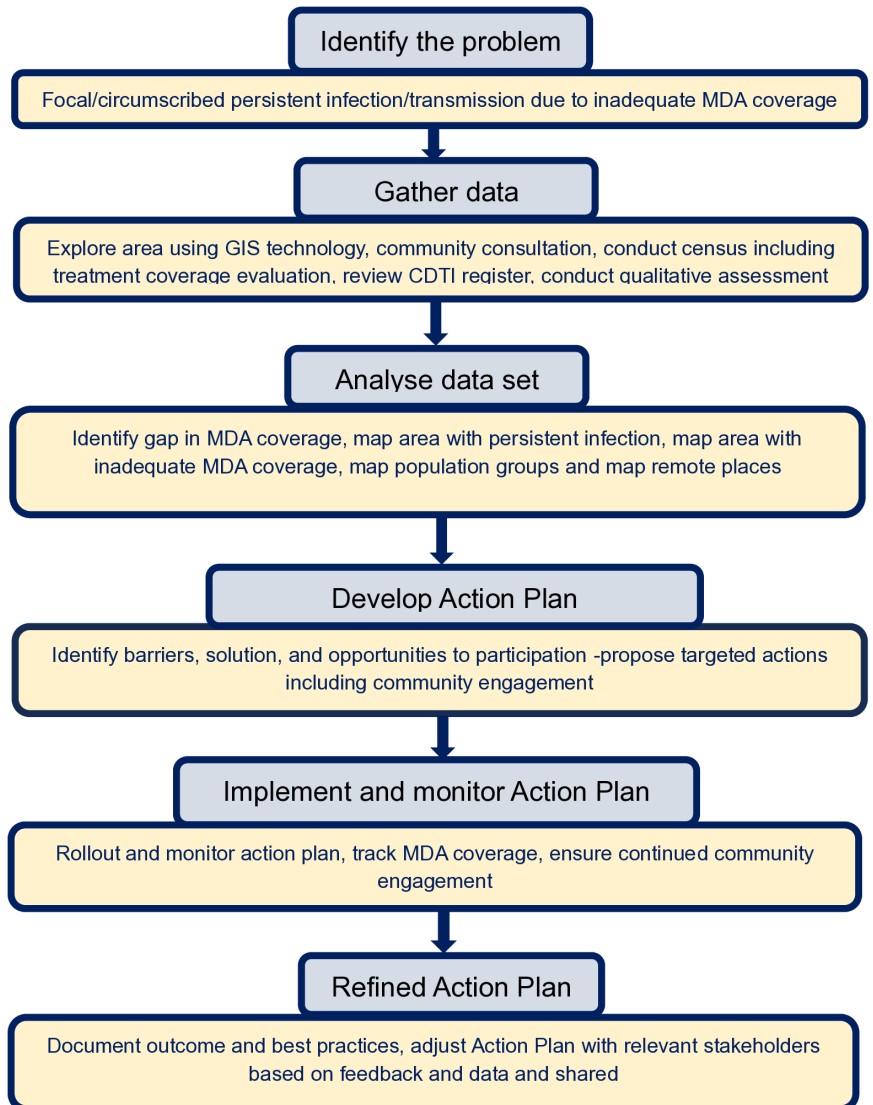

**Fig 2. Study design and procedure.** All steps involved strong community engagement and participation.

**Table 1. Overview of activities and timelines.**

| Activities | Timeframe (2024) | Details/ Participants |
|---|---|---|
| Community-Directed Treatment with Ivermectin (CDTI) | January | Preceding ivermectin CDTI conducted in February in the area |
| Community stakeholders' meetings and identification of settlements | March | Mapped all relevant settlements within and near the 10 km radius to inform operational planning |
| Register Review, Census, and Coverage Evaluation Survey (CES) | March | Assessed existing treatment registers, conducted household census, and surveyed coverage post-CDTI |
| First qualitative assessment via Focus Group Discussions (FGDs) and Key Informant Interviews (KIIs) | May | Held with community members to explore baseline knowledge, attitudes, and barriers to CDTI participation |
| Development of Action Plan (AP) | June | Developed collaboratively with local stakeholders and sub/district health teams using findings from FGDs, KIIs, Census and CES |
| Implementation of AP | July | Delivered to three sub-districts; included all communities within or near the 10 km radius transmission focus area |
| Second FGDs and KIIs | September | Involved Community Drug Distributors (CDD), sub-district health staff, and community members selected from various geographic areas, infection and coverage levels |
| Refinement of the AP | October | Refined AP collaboratively with local stakeholders and sub/district staff using findings from FGDs, KIIs, programme report and field implementation activity log. |

We defined a settlement as the smallest geographic unit where people reside. A community encompasses one or more settlements and integrates cultural, social, and relational dimensions beyond mere physical location [14]. A series of cascaded consultation meetings, starting from districts through to subdistricts, communities, and settlements, were held from 15-18 of March 2024. During these meetings communities and settlements were informed and their consent obtained and dates for community/settlement-level census and qualitative assessments were agreed upon. Neighbouring settlements, including miners, herders, fishermen, and any other groups not initially listed, were inquired about and included. Additionally, satellite images of the area were presented to participants to aid in identification of possible neighbouring settlements.

### Step 2. Identification of settlements from satellite imagery

A suitable satellite image of the area was obtained from Apollo Mapping (https://apollomapping.com/). The image was taken in October 2023 with a resolution of 75 cm, covering 271 square kilometres, representing 86.3% of the 10 km radius area. The image of the remaining area was completed using freely available satellite imagery, albeit at a lower resolution. Structures indicating potential settlements, including mining sites, were identified based on settlement and mining site features from previous studies [11,15]. Ten trained data collectors verified the settlements alongside the census from March 16–24, 2024. GIS data was combined with census data to produce an updated community/settlement map of the 10- km radius area of Kwanware-Ottou onchocerciasis transmission focus.

### Step 3. Census, including treatment coverage surveys

The census team conducted census a month after MDA but before implementation of the AP in all places identified as settlements based on community consultation and field verification of potential settlements identified on satellite imagery. The team also noted neighbouring settlements and evidence of mining activities and associated miners, including them if not already included. All individuals in the area were targeted for enumeration. The following information was collected for everyone: infection status recorded in the 2021 survey [11], demography, occupation, movement, and participation in CDTI, among others (see S1 Text). By covering the entire population, the census ensured a high level of precision for treatment coverage measure.

An initial review of community registers revealed significant quality issues, leading to the decision to abandon the option of updating the existing registers. Challenges included inconsistent naming of individuals, the inclusion of persons who had died or moved away but were still marked as treated, and the continued use of outdated registers for over five years without systematic updates. To address these concerns, the programme opted to conduct a fresh census with (new) paper-based and electronic registers to ensure accurate tracking and follow-up.

**Step 4: Qualitative assessments identifying specific challenges and solutions**

Interview participants were purposefully selected based on their involvement in CDTI activities and their perceived influence within their communities. Additional participants included individuals who self-identified as having never received treatment or only been treated once, infected individuals (based on 2021 skin snip microscopy results), cattle herders—primarily of Fulani ethnic group—miners, and residents of settlements with low coverage or very limited accessibility. This approach enabled the collection of targeted information relevant to programme improvement.

Ten focus group discussions (FGDs), each involving 6–10 participants, and 28 key informant interviews (KIIs) were conducted across 14 communities. In addition, two KIIs were conducted with sub/district programme staff. Trained interviewers used structured guides (S2 Text) to explore the influence of attitudinal factors, mobility, and other barriers on participation in CDTI as well as gathered suggestions on how to improve CDTI delivery and uptake. Facilitators managed group dynamics to prevent dominance by any individual, inviting potentially dominant participants to speak last. Individual interviews in rural communities are often less effective due to participant shyness or fear. Group discussions encouraged open expression, often influenced by peer interaction [16]. On the other hand, community attitudes are shaped more by local leaders' views than by personal opinions [17] hence the need for group and individual interviews.

**Step 5: Participatory development of Action Plan (AP)**

A participatory workshop [18] led by the National NTD programme was conducted in Wenchi on June 5–6, 2024. During the workshop, an AP was developed based on findings from previous steps, 1–4 above. The workshop brought together stakeholders involved in CDTI across the communities selected based on their stake and influence on CDTI. This included frontline health workers (6 individuals), district programme staff (6) national programme staff (4) and representatives from the communities with high infection and/or poor survey treatment coverage (14). Participants reflected on findings, provided feedback, made recommendations and developed actions to be implemented during the following CDTI. A district monitoring team composed of programme staff at the district and subdistrict levels as well as community members was set up to monitor implementation of the actions.

**Step 6: Implementation and Evaluation of Action Plan (AP)**

Actions agreed upon in step 5 were implemented during the subsequent CDTI in July-August 2024. The monitoring team observed processes, noted issues/complaints, and resolved them during distribution of ivermectin. A qualitative assessment of the CDTI was undertaken from 4-7 September using KIIs and FGDs to further evaluate the implementation from the perspectives of community members (see S2 Text for interview guide). Findings from monitoring, qualitative assessment, and programme coverage report were presented in a second workshop held from 9-10 October 2024. This workshop was under the stewardship of National NTD programme (NTDP). Participants mirrored that of AP development and included national programme staff (2 individuals), district NTD focal persons (3), sub-district disease control officers (3), and community members—including Community Drug Distributors (CDDs) and community opinion leaders(6)]. This difference in number from the AP development meeting was due to financial constraints rather than lack of interest or refusal. Participants reflected on the actions implemented, successes, challenges, and agreed on how to improve during subsequent CDTI.

## Data management and analysis

A GIS expert verified completeness and accuracy of GIS locations as they were being collected. Following field collection, data were downloaded and cleaned with confirmed settlements mapped. Census data was downloaded

from the Commcare platform and cleaned and analysed using STATA statistical software (StataCorp. 2023. Stata Statistical Software: Release 18. College Station, TX: StataCorp LLC). The population of each settlement was estimated from the census. Logistic regression was used to estimate the association (odds ratio, OR) of each variable with the outcome of interest (treatment) at 95% confidence interval (CI). $P < 0.05$ was considered statistically significant.

During qualitative data collection, data collectors held daily meetings to discuss and synthesize findings, identify areas for further exploration, and determine points of saturation. Interviews were recorded using digital Dictaphones and later transcribed verbatim. To ensure quality of transcription, research assistants that conducted the interviews also performed the transcriptions [19]. However, all transcripts were independently reviewed and verified by a second researcher to ensure accuracy and reduce interpretive bias. Analysis followed a thematic approach, developed based on participant responses and study objectives, incorporating both deductive and inductive elements to capture emerging insights. NVivo software (version 14) was used to organize and manage the coding process. Verbatim quotes were selected to best illustrate key themes aligned with study objectives. This was used to triangulate quantitative findings from both programme-reported and survey-derived coverage data. Post-implementation programme coverage was compared to pre-intervention survey coverage to assess changes attributable to the intervention package.

## Results

### 1. Identification of settlements through community consultations

During community consultations, 10 hamlets – small remote settlements (Ampiani, Kwaedenden, Bofuorkurom, Kwadom, Kwadom-Nsuta, Kwadom-Boase, Winamda, Branam State Farm, Yibong-Akura, and Mambeele) were identified and included. These had all been included into larger communities during preceding CDTI round.

### 2. Identification of settlements from satellite imagery

The outcomes of the first few days of field verification were useful for data collectors to prioritize potential settlements to visit subsequently. Of 479 locations identified on satellite image, 265 were visited from March 16–24, 2024. Local guides confirmed that remaining unvisited locations were abandoned or temporary shelters. Households were mapped as unique settlements based on their clustering. Combining GIS and census data, 19 named settlements with inhabitants were identified. Fig 3 maps these settlements within the 10 km transmission focus. Of these settlements, 12 did not have own community treatment registers during preceding CDTI which masked the need for CDDs as well as straining their ability to reach them.

### 3. Census and treatment coverage evaluation

A total of 3,338 individuals irrespective of their treatment eligibility were censused in the 19 communities/settlements. In March 24, 70.8% (Confidence Interval, CI: 69.2-72.4) of the total population indicated they swallowed ivermectin in the preceding CDTI carried out 1–2 months earlier. Programme reported a coverage of 88.9% which falls well above the upper limit of the survey coverage's 95% CI. Eight communities, however, reported coverage levels of less than 65%, with the lowest reported in Papaase (25.8%, CI: 13.5-43.8), Kwanware (46.4%, CI: 36.7-56.3), Mensakrum (59.6%, CI: 53.6-65.2), and Gyabaa (46.9%, CI: 33.5-60.8).

A total of 548 eligible participants (20%) were not offered ivermectin. When grouped into main (large and accessible) and satellite (remote with poor accessibility) communities, the difference in the proportion of people not offered ivermectin was significant (15% vs. 24%, respectively; $p < 0.001$). All 466 individuals (14.0% of total population) who reported not registered during the preceding CDTI also indicated that they were neither offered nor took ivermectin, as expected. Thus 85.0% (466/548) of those not offered ivermectin were not registered. Among those not offered ivermectin, 30% (134 out of

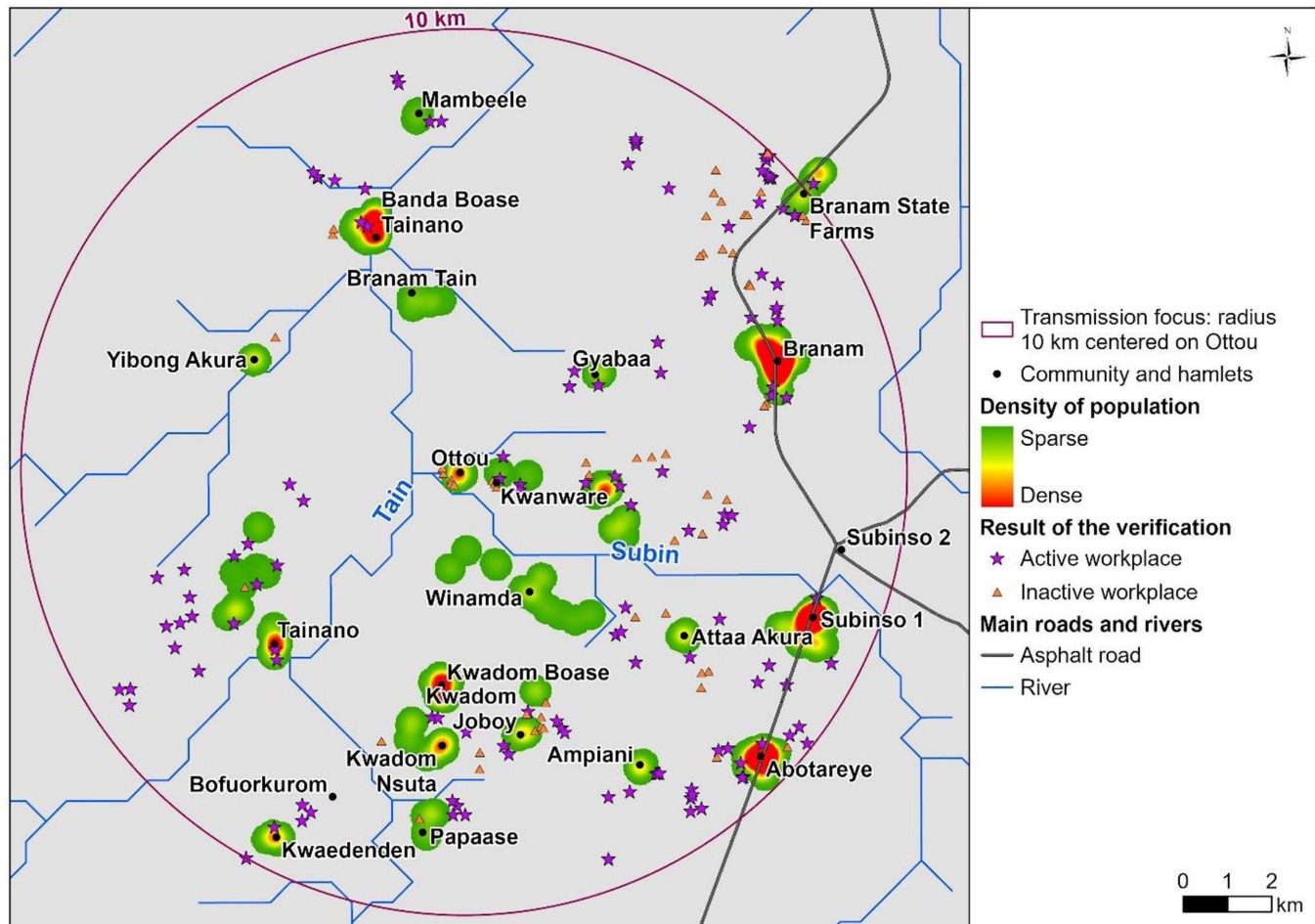

**Fig 3. Map of verified settlements.** The circle indicates a 10 km radius transmission focus, defined based on parasitological, serological, and entomological data. These data show a peak in transmission intensity at the center of the circle, gradually declining to undetectable levels at the 10 km boundary [11]. **Data Sources and References:** The base map uses national and district administrative boundaries from the geoBoundaries Global Database (https://www.geoboundaries.org/countryDownloads.html; license: CC BY 4.0, https://www.geoboundaries.org/metadata.html). Population density and survey results were obtained from primary field-collected data. Additional data sources include HydroATLAS hydrography (https://www.hydrosheds.org/hydroatlas; CC BY 4.0), manually digitized roads derived from high-resolution satellite imagery, and other primary field-collected population and survey data. The map was created in ArcGIS Pro software using only openly licensed datasets.

452 respondents) reported that CDDs did not visit their communities. Eight out of ten individuals who were offered but did not swallow ivermectin cited fear of adverse effects as the reason. Among those who took ivermectin, 3.1% (73 individuals) reported experiencing adverse effects with the most frequent being itches. See Fig 4.

Other reasons for not taking ivermectin included absenteeism and communication issues. Among the 47 individuals who did not take the medicine due to being absent, 22 (46.8%) were at work or school, and 16 (34.0%) were traveling outside the community. Additionally, 167 age-eligible individuals reported not hearing about the MDA. A slightly higher proportion of individuals in satellite communities 96 (6.6%) were unaware of the MDA compared to those in main communities 71 (5.5%; p = 0.23). Regarding satisfaction, 65 participants expressed dissatisfaction: 54 had not received ivermectin, and 7 cited adverse effects. Among those dissatisfied, 75% (49 out of 65) suggested that CDDs should visit them or their communities to offer the medication.

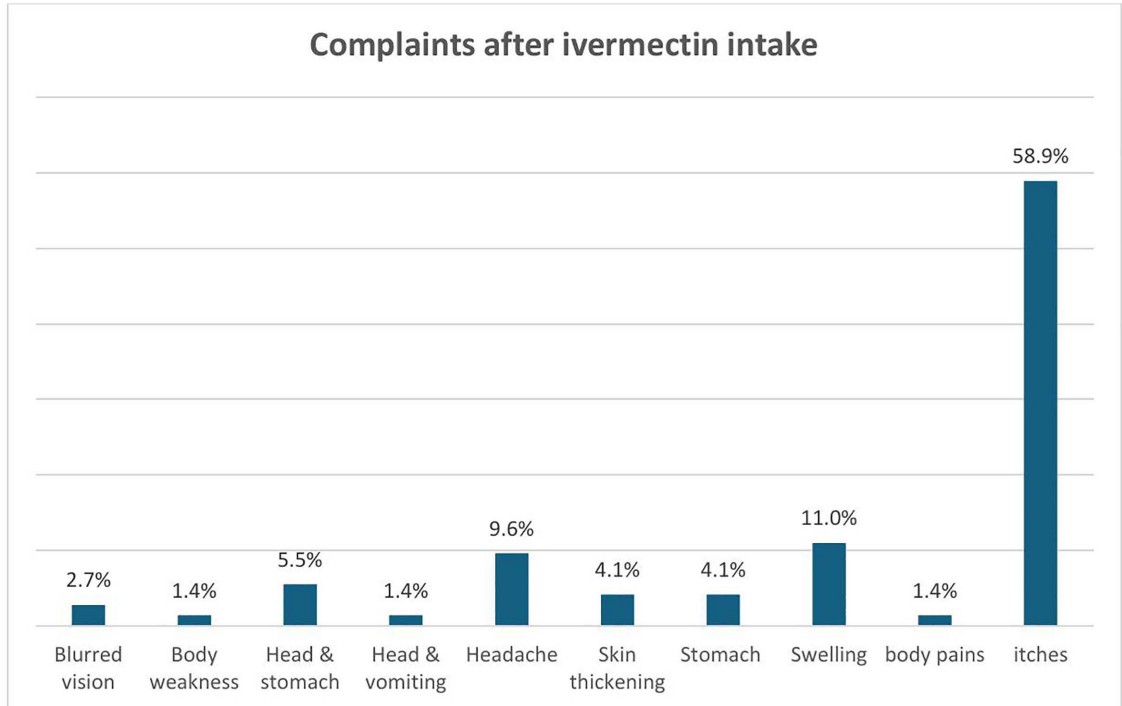

**Fig 4. Complaints after ivermectin intake among 73 respondents.**

In a multiple logistic regression, compared to those whose main occupation was farming, individuals whose main occupation was mining were less likely to have swallowed ivermectin (OR: 0.26; 95% CI: 0.10–0.72). Regarding movement histories, those visiting the community (0.18; CI: 0.08 - 0.40) and permanent residents who had travelled outside the district (0.26; 0.17 – 0.38) were both less likely to have taken IVM compared to permanent residents who had not travelled outside the district in the past twelve months. Similarly, shorter stay in the community was associated with lower OR of taking ivermectin. See Table 2. May, June, and July were reported as months of least movement out of the community.

## 4. Qualitative assessments identified specific challenges and solutions

Key findings from the qualitative assessment revealed context-specific challenges and solutions aimed at improving participation in CDTI. These issues were then considered during development of the AP. Key thematic issues are highlighted below.

### Misconceptions about onchocerciasis and its mode of transmission

Awareness refers to the accurate information that community members have about onchocerciasis and CDTI process—its purpose and benefits—which, influence their engagement and participation [20]. Awareness is influence by misconception. Misconceptions about onchocerciasis transmission among some community members included the beliefs that the disease is gotten from a contaminated water source or mosquito bite.

*"The disease is from the river we drink from the streams; people urinate into the water while animals drink the water too. We need to tackle that first else we can take the drug for thousand years and still get the infection". (FGD with adult men, Wenamda)*

Table 2. Ivermectin Coverage Evaluation by Settlement.

| Communities/ Settlements | Pre-intervention Survey Coverage | | Post intervention Reported Coverage | |
|---|---|---|---|---|
| | Total[1] | Coverage, %, (95% CI) | Total[1] | Coverage, % (95% CI) |
| All | 3338 | 70.8 (69.3–72.4) | 52812 | 88.2 (87.9–88.5) |
| **Predominantly Mining Communities/Settlements** | | | | |
| Banda Boase Tainano[2] | 272 | 59.6 (53.6–65.2) | 242 | 93.8 (90.5–96.7) |
| Mambeele[2] | 1 | 100 | | |
| Yibong Akura[2] | 17 | 64.7 (40.4–83.2) | | |
| Tainano | 184 | 79.9 (73.5–85.1) | 128 | 87.5 (81.2–93.0) |
| **Predominantly Farming Communities/Settlements** | | | | |
| Ampiani | 33 | 90.9 (75.3–97.0) | 76 | 92.1 (85.5–97.4) |
| Branam state farms[2,3] | 68 | 79.4 (68.2–87.4) | 112 | 93.8 (89.3–98.2) |
| Kwadom-Boase[2] | 163 | 65.6 (58.0–72.5) | 192 | 84.9 (79.7–89.6) |
| Kwadom-Joboy[2] | 30 | 76.7 (58.5–88.5) | | |
| Kwadom-Nsuta | 122 | 63.9 (55.1–72.0) | 117 | 82.1 (74.1-88.0) |
| Gyabaa (Branam Tain) | 49 | 46.9 (33.5–60.8) | 52 | 100 (93.1 -100) |
| Papaase[2] | 31 | 25.8 (13.5–43.8) | 61 | 77.0 (65.6–86.9) |
| Kwaedenden (Bofuorkurom)[2] | 31 | 64.5 (46.6–79.1) | | |
| Winamda | 83 | 62.7 (51.8–72.4) | 212 | 90.6 (86.3–94.3) |
| Kwanware | 97 | 46.4 (36.7–56.3) | 107 | 85.0 (77.6–91.6) |
| Ottou | 68 | 86.8 (76.5–93) | 69 | 94.2 (88.4–98.6) |
| Abotareye[2,3] | 470 | 78.9 (75.0–82.4) | 1164 | 85.1 (83.1–87.1) |
| Branam[2,3] | 835 | 69.5 (66.2–72.5) | 1403 | 90.4 (88.9–91.9) |
| Subinso 1[2,3] | 751 | 76.1 (72.9–79.1) | 922 | 88.2 (86.0–90.2) |
| Ata Akura[2] | 33 | 66.7 (49.2–80.5) | | |

[1]Total includes both eligible and ineligible individuals to ivermectin treatment (e.g., children under 5 years of age, pregnant and women breastfeeding <8 days old child, or those with contraindications to ivermectin). [2]These communities/settlements were combined during community-directed treatment with ivermectin (CDTI). [3]Include communities outside focal area. CI: Confidence Interval. Survey conducted in March to evaluate January CDTI; Action Plan implemented in June–July.

As solutions, participants from different communities recommended increasing awareness about the importance of the drug, its possible adverse effects, and how to manage these adverse effects through community meetings. **"**I know we have not had such a gathering since I came to this community 22 years ago." The participants believe it will encourage community participation in the CDTI going forward:

*"I would suggest that we hold a community meeting or 'durbar' where the experts and elders' team up with the agent to educate us. A general meeting would help bring out necessary questions or information to learn from which will help us all." (KII with parent of never treated individual, Branam)*

Participants suggested that the community should be informed of the treatment in advance to ensure people are available during the CDTI.

*"They do not inform that they will be distributing the medicine ahead of time. Inform us ahead of time! We can organize the community for the effective distribution of the medicine." (FGD with men, Kwadom Joe-boy Community).*

*"You could sometimes give the drug to someone, and they refuse to take them because they do not know its use." (KII with CDD, Kwadom Joe-boy Community)*

### High migration of community members

One challenge for implementing CDTI in these communities is the high level of migration during the farming season. CDDs reported that when distribution is during rainy season, they are unable to access some community members who might have moved temporarily to farms. In addition, they stated that some people's nature of work demands that they travel away from the community for a certain period.

*"I am usually absent anytime they come to distribute the drugs. And when they come, they give to available persons only. They do not give to you other people's drugs to keep for them." (KII with never treated person, Branam)*

*"There are some places where the person (referring to the CDD) will tell you that your name is not in their records hence they cannot give you. For me, I haven't taken the medicine for a while, and it is because I travel." (FGD with adult men, Kwadom Nsuta)*

### Low participation of cattle herders

Cattle herders—particularly semi-nomadic individuals from the Fulani ethnic group—reported that they were often absent during MDA campaigns due to their daily mobility and seasonal migration patterns. However, we recognize that not all individuals engaged in cattle rearing are nomadic. The cattle headers occupation category in Table 3 had relatively high coverage (71.7%). This category likely includes settled cattle herders who are more consistently present in the community and therefore more likely to participate in CDTI.

### Lack of resident community drug distributors

Some communities lack resident CDDs.Hence, CDDs from other communities visited to distribute ivermectin. Community members reported that the absence of resident CDDs led to many community members missing treatment as they were absent when the CDD visited their communities. They requested for CDDs to be selected from their communities to ensure everyone is treated during CDTI:

*"The distributor comes to the community during the day when everyone has gone to farm, so many did not get the drug but if the distributor is resident in the community, he will distribute the drug in the evening when everyone will be back from the farm and then no one will be left out" (FGD with Men, Kwadom Nsuta)*

Community members also suggested that the number of CDDs should be increased.

*"So, if you could add more CDDs, it could help to do effective work. When you look at the Boase village and this place and the other surrounding villages, it shouldn't be one person alone sharing for all these villages." (FGD with adult men, Kwadom Nsuta)*

### Lack of training for some Community Drug Distributors

CDDs identified a lack of training as an issue affecting effective delivery of medicine to the community. For instance, one CDD said he "had received no formal training although he had been distributing drugs for five years". He "relied on

**Table 3. Factors associated with participation in Community-Directed Treatment with Ivermectin among participants who were 15 years or older.**

| Characteristics | Categories | Coverage, %[1] (n) | OR[2] (CI) |
|---|---|---|---|
| | All | 80.6% (1467) | |
| Sex | Female | 79.3% (706) | Reference |
| | Male | 81.9% (761) | 1.15 (0.88-1.51) |
| Age | 15-24 | 75.3% (431) | 1.01 (1.00-1.03)[3] |
| | 25-34 | 76.7% (340) | |
| | 35-44 | 82.4% (263) | |
| | 45-54 | 87.2% (184) | |
| | 55 and above | 90.9% (249) | |
| Ethnic group | Akan | 89.5% (212) | Reference |
| | Dagaarba | 78.6% (809) | 0.67 (0.39-1.15) |
| | Fulani | 52.1% (25) | 0.33(0.13 - 0.90) |
| | Mo | 80.4% (115) | 1.03 (0.50 - 2.11) |
| | Others | 84.5% (306) | 1.2 (0.66 - 2.17) |
| Main occupation[4] | Farming | 81.5% (998) | Reference |
| | Cattle rearing | 71.7% (43) | 1.21 (0.60 - 2.42) |
| | Mining | 50.0% (8) | 0.35 (0.12 -1.06) |
| | Sale/office/civil servants | 83.0% (186) | 1.60 (1.04 - 2.46) |
| | Student | 79.2% (164) | 0.99 (0.66 -1.48) |
| | Unemployed | 73.7% (56) | 0.70 (0.38 -1.28) |
| Residency | Permanent - did not travel outside the district | 83.4% (1,389) | Reference |
| | Permanent - travelled outside the district | 60.5% (69) | 0.26 (0.17 – 0.38) |
| | Visitor | 23.1% (9) | 0.18 (0.08 - 0.40) |
| Duration of stay in the community | >10 years | 87.8% (1,081) | Reference |
| | 0-5 years | 57.1% (204) | 0.23 (0.17 - 0.31) |
| | 6-10 years | 78.8% (182) | 0.63 (0.44 - 0.91) |
| Education | Yes | 80.7% (849) | Reference |
| | No | 80.6% (618) | 1.04 (0.79 - 1.39) |
| Sub-District | Boase | 74.3% (130) | Reference |
| | Nsawkaw | 86.4% (108) | 2.55 (1.35 - 4.80) |
| | Subinso | 80.9% (1,229) | 1.27 (0.83 - 1.94) |
| Community | Main | 83.1% (857) | Reference |
| | Satellites | 77.4% (610) | 0.70 (0.51 - 0.95) |

[1]Odds ratio. [2]The percentage represent the proportion of individuals within each subgroup who took ivermectin along with the actual number in parenthesis. [3]Age was treated as a continuous variable. [4]Occupation data was missing for 12 participants. Only participants aged 15 years or older were included to ensure they had been age-eligible for ivermectin treatment for at least 10 years, starting from the age of 5. This criterion was applied to ensure that age did not exclude individuals from any category of variables.

personal discretion to select who was offered the drug and the dosage". This concern was also identified by a community head in another community, reporting that CDDs "no longer measured the height of individuals before distributing ivermectin", because of an absence of training. He raised concern that dosages could be incorrect.

*"Initially they used to measure us and so from there, he knows that I am X inch and so he must give me four [tablets]. But now, he uses his eyes to do it." (KII with Community Head, Kwanware)*

**Directly observation therapy (DOT) not practised**

Another challenge of implementing CDTI identified in some communities is that DOT, recommended by the programme, is not strictly adhere to. Community members reported that CDDs distributed the drugs without ensuring it was swallowed, yet noting the individual as treated in the register:

> *"Initially when they started the distribution, they ensured that people took the drug in front of them [CDDs], but now they have stopped. So, it makes it difficult to know those who have actually swallowed the drug." (FGD with men, Wenamda).*

> *"Some people collected the drug and did not swallow it but kept it in their rooms and you won't know." (FGD with adult men, Kwadom Nsuta)*

Community members therefore recommended that DOT should be practiced. In doing this, persons who do not take or refuse to swallow the drug would be easily identified for action.

**Refusal to take the medicine due to adverse effect**

A major reason given by participants for low participation in CDTI in the communities is the fear of adverse effects after actual experience or observation in others who swallowed the medicine. Women, mostly from Kwadom Nsuta settlements, narrated their ordeal after taking the medicine and decided they would not take again.

> *"About three years ago, I was given oncho [onchocerciasis] drug [ivermectin], and my body got swollen. I when to the hospital several times. I was given injection, but the condition was the same. The condition is no more with me but, if they bring the oncho drug again, I will not take." (FGD with women, Kwadom Nsuta)*

> *"No…. when they brought the drug, I too, did not take. As for oncho truly speaking, I nearly died, so, I will not take again." (FGD with women, Kwadom Nsuta.)*

**5. Participatory development and implementation of Action Plan (AP)**

The AP development workshop was highly participatory and interactive, engaging a broad range of stakeholders including health administrators, District NTD Focal Persons, Sub-district representatives, and community leaders and members. All participants attended and actively contributed. CDDs appreciated the involvement of opinion leaders and local health authorities in jointly identifying solutions. They noted that this collaborative engagement was the first time and that the approach mobilized effectively community and changed community's perceptions of CDTI from being solely a CDD responsibility to that of the whole community. Community members valued the opportunity to have their voices heard and considered in shaping CDTI activities.

The AP specified activities and the responsible parties for each activity, outlined reasonable timelines, and identified the resources and capacities needed for implementation. Key components of the AP included strengthening local advocacy and resource mobilisation, engaging with hard-to-reach groups including semi-nomadic cattle herders and artisanal miners, conducting community outreach to raise awareness, shifting CDTI period to July (peak immigration), and recruiting and (re)training all CDDs on awareness raising, census and (re) registration of community members using new paper register and electronic platform as well as ivermectin administration using dosing poles. Although community members expressed the need for additional CDDs, the final number was determined by the programme during AP development, based on feasibility and operational needs. This decision considered factors such as population size, settlement patterns, accessibility and resources. During the implementation of the AP, field experiences and adjustments were recorded in an

activity log. Summary of the AP implementation process is found in S3 Text and the district refined AP are found in S4-S6 Tables.

## 6. Evaluation of action plan implementation

This was achieved through both qualitative and quantitative assessments.

### 6.1. Qualitative assessment of action plan implementation

Participants indicated that significantly more people were treated compared to the previous CDTI round. They appreciated the various actions taken that encouraged greater community participation.

*"Yes, they came to inform us that they would be bringing the drugs. This time, there was someone in the community [a newly recruited CDD], and he made sure everyone who was eligible to take medicine got it."*

*"Yes, there was enough medicine for everyone. I did not hear of any medicine shortage." (FGD with men, Kwadom Nsuta)*

Furthermore, participants appreciated the timing of the MDA, stating that it corresponded to when people were available.

*"I also like the time. Blackflies bring the oncho disease, so during the rainy seasons, that is when the flies are dominant and people are around, so this was the same time they brought the oncho drugs." (FGD with women, Kwadom Boase).*

Lastly, the acceptability of the programme had improved among the community members based on their new understanding of the importance and benefits of the medicine.

*"Initially, community members were a bit hesitant to take the medication, but this time, when it came, people came for it in a rush. This time, almost everyone, both children and adults, was aware of the medication." (KII CDD, Ottou)*

Participants also highlighted areas for further improvement. Notably, the request for a follow-up to treat those who were missed.

*"When they left, some people came saying that they did not get the drugs. So, I am pleading that as you go, tell them not to delay coming and share for those who did not get it. When they came, their wives and children were not around. So, they should come back and give them the drug". ((FGD Kwadom Boase men)*

### 6.2. Quantitative assessment of action plan implementation

The sub-districts have well-defined geographical boundaries and serve as the primary level for CDTI programme data collation. Therefore, we consider this suitable geographic scale for evaluating the impact of the intervention. For the three sub-districts included in the intervention, the old registers recorded 6,898 more individuals. This is 13.1% higher than the new register record (59,710 vs. 52,812) even though the new registration was during the peak immigration (rainy) season. Since the intervention was closely monitored, the treatment coverage data obtained was reliable and comparable to the survey coverage recorded during the initial census. For the population in mining settlements which were located entirely in the focal area, a smaller population was counted during census compared to the new registration - 474 vs 370— 21.9% less. Conversely, farming settlements saw an increase from 707 to 886, 25.3% higher. Coverage increased by 17.4% (70.8% vs 88.2%). All communities/settlements achieved and exceeded the 65% recommended coverage, indicating a substantial improvement across the board. See Table 2 above.

## 7. Refinements of the action plan

Field experiences documented in the activity log, along with qualitative assessments and quantitative analysis of reported treatment coverage across the three sub-districts, informed the refinement of the AP. The refinement was carried out during a workshop involving programme implementers and community stakeholders. They reflected on successes, challenges, and opportunities for improvement. Key challenges identified included limited interest among youth in serving as CDDs, short CDTI periods, delays in fund transfers from national to district levels, high mobility of miners and nomadic cattle herders, and language barriers—particularly with French-speaking artisanal miners and some herders. These challenges and solutions were reflected in the AP. Despite the operational difficulties, programme implementers commended the AP development and implementation as a viable and adaptive approach for continuously addressing field-level issues and enhancing programme delivery. Further details are provided in S3 Text & S4-S6 Tables.

## Discussion

This study piloted a comprehensive, mixed-methods approach to improve CDTI performance in a persistent onchocerciasis transmission focus in Ghana. It involved sequential steps including community consultation, GIS mapping, register review, census and treatment coverage evaluation survey, qualitative assessments, and participatory action planning. The intervention identified operational gaps and implemented targeted context-specific solutions. This implementation process was successfully embedded within national health structures including alignment with sub-district health units. Community trust and ownership were fostered through continuous engagement, tailored awareness campaign, and the integration of local perspectives. The resulting AP emphasized local solutions, resource mobilization, and accountability. The AP implementation was effective, built capacity at all levels, and generated valuable insights to inform future efforts.

Prior to the intervention, data quality was poor, characterized by significant overreporting while simultaneously missing some individuals. Overreporting was largely due to outdated registers and cumulative registration over more than five years, which included individuals who had migrated or died but were still listed and marked as treated. Misreporting could have occurred in five key ways: 1 individuals no longer residing in the community (died or moved away) but still listed and recorded as treated; 2 individuals absent during drug distribution but likely marked as treated; 3 individuals present in the community but were not reached by CDDs yet recorded as treated;4 individuals who were offered treatment but did not swallow the medication but recorded as treated. This stemmed from the absence of DOT, which was in turn challenged by remote communities and CDD-related constraints; 5 lastly, individuals present but not registered and not treated. Although this group had no impact on register-based coverage estimates, they pose a threat of perpetuating transmission. Eighty-five percent of those not offered treatment also reported not being registered. All these contributed to inflated population figures and treatment coverage. Despite being conducted during peak immigration season, the new registers recorded almost 7000 (6,898) fewer individuals than in the old registers. This illustrates both significant population turnover over more than five years of cumulative registration and persistent data quality issues. Preceding CDTI reported coverage (88.9%) significantly exceeded survey coverage (70.8%; 95% CI: 69.2–72.4), indicating overreporting besides a substantial number of missed individuals (466) in register. This effectively demonstrates data issues which may have led to resource wastage (e.g., drugs) and misleading assessments of adequate coverage. However, only a comprehensive review of the old registers couple with census can quantitatively reveal the full extent of the misreporting issues.

Specific subgroup challenges were identified, notably among unauthorised mining, farming, and semi-nomadic cattle-rearing communities. For farming communities entirely within the focal area, there was a 25.3% higher population in July compared to March. This is attributable to seasonal immigration during the rainy season for agricultural activities [21,22]. In contrast, the 21.9% lower population observed in mining communities reflects seasonal emigration of miners [23]. This suggests that CDTI efforts should be reinforced in mining communities during the dry season, while in farming communities, they should be intensified during the rainy season. The Biannual CDTI being implemented in the area [9] should align with this pattern. Semi-nomadic populations, in addition to their mobility, often face conflict with host

communities due to the destruction of crops by their livestock. This has fostered mistrust, subsequently affecting their uptake of host community-driven CDTI initiatives [24]. These observations highlight complex population dynamics in the Kwanware-Ottou focus, contributing to persistent onchocerciasis transmission.

The implementation of AP has improved data accuracy and quality and enhanced both geographic and therapeutic coverage. Post-intervention coverage was reliable, following the introduction of both paper-based and electronic registration. DOT was reinstated, supported by intensified field supervision and monitoring. These enhancements made post-intervention coverage comparable to survey coverage. Therapeutic coverage significantly improved across all communities and settlements, surpassing 80% recommended threshold [25,26] in all but one location. Overall coverage increased markedly, rising from 70.8% (95% CI: 69.2–72.4) to 88.2% (95% CI: 87.9–88.5)—a gain of 17.4%. Uptake from hard-reach communities where persistent infection is feared, likely contributed more to this increase. These results highlight the effectiveness of targeted, data-driven interventions in enhancing programme performance and promoting more equitable access to treatment.

This study builds on and extends existing evidence on strategies to improve CDTI performance for onchocerciasis. In Benin, the use of rapid ethnography and participatory techniques led to a 13% increase in CDTI coverage by addressing community trust, supervision, and messaging challenges. However, that approach did not incorporate spatial mapping or structural reforms such as register validation [27]. Similarly, studies in Ghana's Atwima Nwabiagya North District and in Benin identified low awareness and poor communication as key barriers to CDTI uptake, underscoring the importance of targeted awareness raising strategies [28]. In Côte d'Ivoire and Uganda, interventions focused on strengthening the motivation and resilience of CDDs, emphasizing community support [29] but not addressing systemic data quality or operational restructuring as in this study.

In contrast, this study uniquely integrates GIS mapping, community register review, census validation, and participatory planning within a clearly defined transmission focus, executed through a seamless and coordinated rollout. This comprehensive, systems-oriented approach—embedded within national programmatic structures contributed to the higher observed increase in therapeutic coverage - 13% in Benin [27] vs 17 in this study. This is a scalable model for improving CDTI outcomes in complex and underserved settings. Moreover, it demonstrates how such integrated strategies can be effectively incorporated into routine programmatic activities such as CDD training, community mobilization, and microplanning. The outcomes provide a more accurate and dependable foundation for future monitoring, evaluation, and accountability.

There are notable limitations in this study to consider. Qualitative methods like FGDs are prone to bias stemming from group dynamics and dominant voices, which may limit their reliability in capturing individual-level changes. This was addressed by training interviewees who ensured that all participants contribute to the discussion. Transcription bias may have occurred, as interviewers also transcribed the data, though this was mitigated through independent review. However, future studies should use standardized comparative surveys, alongside qualitative assessment, to strengthen the evidence for impact. No formal audit of CDD deployment was conducted, limiting insights into workforce gaps. Decisions regarding additional recruitment and placement were made collaboratively with district health teams based on feasibility and operational needs. To our knowledge, there is no nationally standardized protocol for determining the number or distribution of CDDs in onchocerciasis-endemic settings in Ghana. The required number of CDDs is often conflated with the number of community-based surveillance (CBS) workers, who are recommended at a ratio of one CBS worker per 500 people [30]. Indeed, program data cited by Amazigo et al. (2021) indicated a ratio of one CDD per 568 people in Ghana [31], significantly exceeding the WHO recommendation of one CDD per 100 people. However, this WHO benchmark is context-dependent and not universally adopted [31]. We acknowledge this limitation and recognize the value of CDD workforce audit and need to establish structured guidelines to support future programmatic planning and equitable distribution of CDDs. While the enhancement approach in this study appears cost-effective—especially in high-transmission areas within a 10 km radius—a formal costing study is needed. This echoes concerns raised in the Benin study [27],

where stakeholders worried about the cost of scalability of participatory ethnographic methods. However, the AP tools developed here can be integrated into the programme at no additional cost for several years, unless new alerts arise.

Together with our earlier work [11], this study highlights three foundational and sequential pillars for holistic investigation and intervention to improved CDTI:

1. **Investigation and Definition of Scope**: Start by investigating programme alerts to identify evidence of ongoing transmission. Employ GIS mapping—focused on rivers and settlements—alongside environmental, entomological, and epidemiological assessments. These define the extent and severity of the problem. Once these assessments are completed, align findings with the smallest health system implementation units, ensuring that all associated communities within the unit are mapped and entirely included. Subsequent intervention steps should correspond directly to these operational units.

2. **Assessment of Treatment Adequacy**: conduct a complete census matching with community registers. Ensure that all communities and their respective CDDs are accurately accounted for. Collaborate with district teams to identify and engage key stakeholders, with particular attention to underserved, infected, and at-risk populations. Conduct census, CES, and qualitative assessment to explore community norms, values, leadership structures, and engagement dynamics specific to different subgroups. Use qualitative methods to elucidate barriers to access, community-driven solutions, and identify practical entry points and approach for improved participation. Emphasize a bottom-up approach to amplify community voices and strengthen local action, including advocacy and resource mobilization. These activities should begin within three months after the CDTI and be completed at least two months before the next CDTI. This allows sufficient time for data collection and analysis, stakeholder engagement, and the integration of findings into the final CDTI strategy.

3. **Bottom-Up Integration of Solutions**: Translate the gathered evidence into feasible and actionable programmatic tools (AP), led by communities and guided by district authorities, with support from national staff. Ensure strong local ownership and alignment with existing health system structures.

These pillars are justified by their alignment with the shifting epidemiology of onchocerciasis, as well as with the structures of health systems and communities. This alignment supports broader scalability and enhances the effectiveness of interventions beyond CDTI, including complementary and alternative strategies.

## Conclusion

This study demonstrates that a scalable community-based stimulus package, implemented through a coordinated approach, is effective. It addresses the evolving epidemiology of onchocerciasis, where transmission now persists in small, localized pockets. This aligns with the goals of the 2021–2030 NTD Roadmap. The approach is applicable not only in Ghana but also in similar endemic settings. As countries progress toward elimination, flexible and locally adaptable strategies like this one will be essential to overcome operational barriers and sustain momentum.

## Supporting information

**S1 Text. Census questionnaire.**
(DOCX)

**S2 Text. Qualitative assessment guides.**
(DOCX)

**S3 Text. Findings, accompanying actions, successes, and challenges of implementation.**
(DOCX)

**S1 Table. Banda district action plan.**
(XLSX)

**S2 Table. Tain district action plan.**
(XLSX)

**S3 Table. Wenchi district action plan.**
(XLSX)

## Acknowledgments

We are grateful to all the participants who willingly took part in this study. Their cooperation and contributions were invaluable. We also extend our sincere appreciation to the programme staff at the National, Regional, District, and Subdistrict levels for their unwavering support and commitment throughout the study. Additionally, we acknowledge the efforts of the community leaders and local stakeholders who facilitated consultations and mobilization, ensuring the smooth execution of the research.

## Author contributions

**Conceptualization:** Rogers Nditanchou, David Agyemang, Mike Yaw Osei-Atweneboana, Richard Selby, Joseph Opare, Louise Hamill, Joseph Nelson Siewe Fodjo, Veronique Verhoeven, Elena Schmidt, Robert Colebunders.

**Data curation:** Rogers Nditanchou, Akinola Stephen Oluwole.

**Formal analysis:** Rogers Nditanchou, Akinola Stephen Oluwole.

**Funding acquisition:** Rogers Nditanchou, Richard Selby, Elena Schmidt.

**Investigation:** Rogers Nditanchou, Akinola Stephen Oluwole, Sapana Basnet, Alexandre Chailloux, Judith Saare, Mike Yaw Osei-Atweneboana.

**Methodology:** Rogers Nditanchou, Sapana Basnet, Alexandre Chailloux, Mike Yaw Osei-Atweneboana, Joseph Opare, Louise Hamill, Joseph Nelson Siewe Fodjo, Veronique Verhoeven, Robert Colebunders.

**Project administration:** Rogers Nditanchou, Judith Saare, David Agyemang, Sandra Adelaide King, Richard Selby, Elena Schmidt.

**Resources:** Rogers Nditanchou, Mike Yaw Osei-Atweneboana, Richard Selby, Elena Schmidt.

**Software:** Rogers Nditanchou.

**Supervision:** Rogers Nditanchou, Akinola Stephen Oluwole, Alexandre Chailloux, Judith Saare, David Agyemang, Sandra Adelaide King, Richard Selby, Joseph Opare, Joseph Nelson Siewe Fodjo, Veronique Verhoeven, Robert Colebunders.

**Validation:** Rogers Nditanchou, Akinola Stephen Oluwole, Sapana Basnet.

**Visualization:** Rogers Nditanchou.

**Writing – original draft:** Rogers Nditanchou, Akinola Stephen Oluwole.

**Writing – review & editing:** Rogers Nditanchou, Akinola Stephen Oluwole, Sapana Basnet, Alexandre Chailloux, Sandra Adelaide King, Mike Yaw Osei-Atweneboana, Richard Selby, Joseph Opare, Louise Hamill, Joseph Nelson Siewe Fodjo, Veronique Verhoeven, Elena Schmidt, Robert Colebunders.

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
