## [Decision Letter · Decision Letter 0]

2 Jul 2025

PNTD-D-25-00747

Reaching the Last Mile with Ivermectin Mass Drug Administration against Onchocerciasis: The case of Kwanware-Ottou persistent transmission focus in the Wenchi Health District of Ghana

Dear Dr. NDITANCHOU,

Thank you for submitting your manuscript to PLOS Neglected Tropical Diseases. After careful consideration, we feel that it has merit but does not fully meet PLOS Neglected Tropical Diseases's publication criteria as it currently stands. Therefore, we invite you to submit a revised version of the manuscript that addresses the points raised during the review process.

*Please note that both reviewers have raised substantial concerns. I agree with them. Please make sure that those concerns are carefully addressed when submitted the revised version of your manuscript.*

Please submit your revised manuscript within 60 days Aug 31 2025 11:59PM. If you will need more time than this to complete your revisions, please reply to this message or contact the journal office at plosntds@plos.org. Please include the following items when submitting your revised manuscript:

We look forward to receiving your revised manuscript.

Kind regards,

Vito Colella, DVM, PhD

Academic Editor

Eva Clark

Section Editor

Shaden Kamhawi

co-Editor-in-Chief

Paul Brindley

co-Editor-in-Chief

**Journal Requirements:**

At this stage, the following Authors/Authors require contributions: Rogers NDITANCHOU, Akinola Stephen Oluwole, Sapana Basnet, Alexandre Chailloux, Judith Sare, David Agyemang, Sandra Adelaide King, Mike Osei-Atweneboana, Richard Selby, Joseph Opare, Louise Hamill, Joseph Nelson Siewe Fodjo, Veronique Verhoeven, Elena Schmidt, and Robert Colebunders. Please ensure that the full contributions of each author are acknowledged in the "Add/Edit/Remove Authors" section of our submission form.

4) We do not publish any copyright or trademark symbols that usually accompany proprietary names, eg ©,  ®, or TM  (e.g. next to drug or reagent names). Therefore please remove all instances of trademark/copyright symbols throughout the text, including:

- © on page: 14

- ® on page: 14.

5) Please upload all main figures as separate Figure files in .tif or .eps format. For more information about how to convert and format your figure files please see our guidelines:

Potential Copyright Issues:

i) Figures 1, and 3. Please (a) provide a direct link to the base layer of the map (i.e., the country or region border shape) and ensure this is also included in the figure legend; and (b) provide a link to the terms of use / license information for the base layer image or shapefile. We cannot publish proprietary or copyrighted maps (e.g. Google Maps, Mapquest) and the terms of use for your map base layer must be compatible with our CC BY 4.0 license.

7) We note that your Data Availability Statement is currently as follows: "All relevant data are within the manuscript and its Supporting Information files". Please confirm at this time whether or not your submission contains all raw data required to replicate the results of your study. Authors must share the “minimal data set” for their submission. PLOS defines the minimal data set to consist of the data required to replicate all study findings reported in the article, as well as related metadata and methods (https://journals.plos.org/plosone/s/data-availability#loc-minimal-data-set-definition).

8) Please amend your detailed Financial Disclosure statement. This is published with the article. It must therefore be completed in full sentences and contain the exact wording you wish to be published.

9) Your current Financial Disclosure states, "Yes ↳ Please add funding details. This work received financial support from the Coalition for Operational Research on Neglected Tropical Diseases (COR-NTD), which is funded at The Task Force for Global Health primarily by the Bill & Melinda Gates Foundation (OPP1190754) and by the United States Agency for International Development through its Neglected Tropical Diseases Program. Under the grant conditions of the Foundation, a Creative Commons Attribution 4.0 Generic License has already been assigned to the Author Accepted Manuscript version that might arise from this submission. Grant No 274G. ↳ Please select the country of your main research funder (please select carefully as in some cases this is used in fee calculation). UNITED STATES - US".

However, your funding information on the submission form is missing details regarding any funds from the Bill & Melinda Gates Foundation and the United States Agency for International Development (USAID).

Please indicate by return email the full and correct funding information for your study and confirm the order in which funding contributions should appear. Please be sure to indicate whether the funders played any role in the study design, data collection and analysis, decision to publish, or preparation of the manuscript.

**Reviewers' Comments:**

Reviewer's Responses to Questions

**Key Review Criteria Required for Acceptance?**

**Methods**

-Are the objectives of the study clearly articulated with a clear testable hypothesis stated?

-Is the study design appropriate to address the stated objectives?

-Is the population clearly described and appropriate for the hypothesis being tested?

-Is the sample size sufficient to ensure adequate power to address the hypothesis being tested?

-Were correct statistical analysis used to support conclusions?

-Are there concerns about ethical or regulatory requirements being met?

Reviewer #1: Objectives of the study are clearly articulated; population is clearly described; ethical requirements were met.

One problem is that the assessment of efficacy was entirely qualitative. The standard in the field is to do comparative surveys, where people are asked to respond to the same series of questions so that comparisons can be made across time points in a statistically rigorous manner, and allowing for assessment of sample size relative to population size etc. Focus groups are not a good method for assessing changes in attitudes, because people's responses in groups are impacted by what is expressed by others around them, and because one speaker could monopolize the response time and their opinions become overly reflected in the resulting transcripts.

The authors clearly state that their assessment is qualitative, and this study is now over so I'm not necessarily suggesting that surveys be done now. However, one of the main goals of the study is to provide an approach for developing action plans that could be applied elsewhere, and assessment of efficacy should be an important component of any action plan. Methods for assessment need to be rigorous and applied both before and after the implementation of the Action Plan. It looks to me like the Census included some parts of a knowledge/attitude/practice survey, in that they asked why people didn't take the drug and whether they knew about MDA. Could the CDDs do phone-based surveys as part of their duties, for example, or is this placing too much of a burden on training?

Why was 10km chosen as a radius? Is there evidence that this range is useful or relevant to oncho transmission (i.e., a transmission zone) or is it arbitrary? Looking at Figure 3, there are communities that appear to be on the border/overlap with the 10km radius.

Having the same person run the interviews and do the transcriptions could introduce errors as there is no independent evaluation. I'm not a social scientist but perhaps check best practices in the field for this.

Reviewer #2: The Methods are adequately described.

**Results**

-Does the analysis presented match the analysis plan?

-Are the results clearly and completely presented?

-Are the figures (Tables, Images) of sufficient quality for clarity?

Reviewer #1: Need to indicate the time scale of each step--when was MDA? when did the first FGD occur? When did meeting with other stakeholders occur? When was the plan developed and who was in the room when that happened? When was the plan implemented? When did MDA occur? Who assessed reported coverage? When were the second FGD/KIIs held and how was it decided who to interview?

Could the quantitative results of why people didn't take the drug be reported in this section?

Refinements: who were the stakeholders involved in this workshop? How many people attended relative to the earlier workshop?

How determine how many additional CDDs are needed and where they should be located? Is there an existing protocol for this that is implemented in Ghana?

Appendix 3 is not the Action Plan, but a summary table that indicates the actions taken and by whom. But the Action Plan itself was meant to indicate who was responsible for what action and how implementation success would be assessed? It would be good to see that document.

Reviewer #2: The authors describe this as a mixed methods study utilising a combination of qualitative and quantitative analyses. My reading of the Results section of the manuscript is that there are almost no quantitative data at all.

For example

• in the section "A review of community treatment registers" there are qualitative statements such as "the number of people included in registers was consistently higher than the numbers counted in the census” but there is no quantitative estimate of how much higher, nor of what “consistently” means. Consistently between communities? Consistently between occupations? Genders? All of which are shown later to be associated with coverage. It is also worth noting that overestimating the number of people eligible for treatment would likely result in an underestimate of true coverage. Given that therapeutic coverage is a key parameter, the scale of the effect that the “consistent” overestimate of the size of the eligible population may hve on coverage reporting should be explored.

• Table 1 reports that 3,338 people were censused in 19 communities. Does “people” refer here to all persons (irrespective of MDA eligibility status) or does it refer only to those people deemed to be eligible for treatment? And how does this number compare with the inflated estimates described qualitatively in the section above? More importantly, the therapeutic coverage for each community in March 2024 is given but there are no data for earlier MDA rounds, so there is no way of knowing whether, and in which direction, there is a change in MDA coverage. To add to this uncertainty concerning how these data are to be interpreted, the text above the table refers to MDA in January 2024 whereas the Table caption refers to March 2024 and cites the same overall coverage (70.8%). So, do the text and the table report the same MDA round? If so, then when did this round actually take place?

• Still on the question of the January/March 2024 coverage data, how does the reported overall coverage of 70.8% compare with the previous round(s)? These data are reported as people who “indicated they swallowed ivermectin in the recent round”. These data are therefore “survey data” i.e. data on coverage collected by asking people directly whether they were treated or not after the MDA round. This is in contrast to the earlier statements referring to treatment records from register data. It is well known that reported MDA coverage from community treatment registers almost always overestimates survey coverage which is measured by questioning people directly.

• Taken together, my comments above point to at least three potential problems with the reporting of coverage in this section. First, the unknown extent of overestimation of the eligible population in the old inaccurate treatment registers should lead to an underestimate of coverage calculated from these registers. Second, recording dead or absent people (in the old registers) as being treated would have the opposite outcome i.e. inflating reported coverage. So, which of these opposing features of the historical coverage estimates is important? What, for example is the effect on the historical reported coverage of simply removing the dead and absent from the registers? And third, the “current” coverage data reported in the table cannot be compared directly with the historic coverage estimates because they are “survey coverage” data derived from direct questioning of eligible people whereas the historic data are “reported coverage” from treatment records. These two methods of estimating coverage almost invariably generate different coverage estimates. My view is that we can conclude nothing concerning the impact of measures taken to improve coverage from the coverage data that are in the manuscript.

The next section of the manuscript deals with qualitative assessment of challenges and solutions. The authors state in the Methods that they used the Nvivo package to analyse the qualitative responses collected from focus groups and discussions. I am not a social scientist and have not used the Nvivo package, but the description of the package states clearly that “offers tools for creating visual representations of data, such as word clouds, concept maps, and charts, facilitating data interpretation and presentation”. This section of the Results makes no reference at all the Nvivo analysis, nor does it contain any of the outputs and data display options that can be generated by the software as aids to more objective, semi-quantitative summaries of the discussions of focus groups etc. What are included in the manuscript are a series of verbatim quotes from focus group discussion participants. How representative are these isolated quotes of the discussion in general? Why did the authors include these specifically? What criteria were used to select these quotes? Are all the transcripts of all the discussions (i.e. the “raw data”) available? If so, where? If not, why not? Other questions that come to mind with respect to this section are:

• the authors state that cattle herders rarely take MDA. However, Table 2 includes “cattle rearing” as an occupation that has a 70.3% MDA rate (compared with 70.8% for the whole population). The basis for the statement that cattle herders rarely participate appears to be the result of assuming that all cattle herders are members of the Fulani ethnic group. Where is the evidence for this?

• lack of community drug distributers: there are recommendations for the ratio of CDD’s per community and/or head of population. What are the numbers in these communities? How far below the recommended number of CDD’s for this size of target population. Likewise, there are guidelines for training frequency and content. What are the training records for CDD’s in these communities? What does lack of training for some CDD’s mean? What proportion of CDD’s do not receive training i.e. what does some mean? Is it a sufficiently large proportion that it is likely to impact MDA? The qualitative statements made here should be able to be supported by quantitative data on CDD numbers, rates of turnover and training that can be related to MDA differences between communities. The qualitative descriptions and anecdotal quotations in the manuscript are of little value in identifying which components of MDA delivery are most likely to contribute to poor outcomes.

Lastly a couple of questions about Table 2.

First, what are the numbers in parentheses? I assume it is the number of people in each category. However, if this is the case and the category “All” 80.3% (1467) means there were 1467 people 15 years and older, why is the category “Permanent resident - did not travel 1603? What is the denominator for the percentage calculations? Again, in the Residency category, “Permanent resident – did not travel is 83.8% (1603) but “Permanent Resident – travelled is 55.9% (76). I cannot make sense of these numbers. Could the authors please explain how the percentages are calculated and what the number in parenthesis after each percentage means?

Second, the footnote to the table states that only people over 15 years old were included to “ensure a fair comparison across of variables”. What does this mean?

I have similar objections to the entirely anecdotal, qualitative way in which the implementation, evaluation and refinement of the action plan are described. The most obvious measure of the success of a plan for improvement would be an increase in therapeutic coverage following implementation of the plan. No such data are included. Without these data it is impossible to draw any conclusions as to whether the barriers to MDA implementation were identified correctly nor whether the remedial actions were effective (nor, indeed, whether this qualitative approach to improving MDA is itself effective).

**Conclusions**

-Are the conclusions supported by the data presented?

-Are the limitations of analysis clearly described?

-Do the authors discuss how these data can be helpful to advance our understanding of the topic under study?

-Is public health relevance addressed?

Reviewer #1: This section was not very well organized, with sentences not really following from the sentence that came before it. Paragraph 2, for example, goes back and forth between topics; paragraph 3 was confusing, moving between treatment coverage, assessments, and uptake without making clear logical links. The discussion about the involvement of the NTD programme seemed like it should have come earlier?

I would encourage the authors to consider the discussion less as a reporting of the results from this region of Ghana, and more as making a targeted recommendation for how an Action Plan can be developed in conjunction with community and NTD programme people based on the experiences of this pilot. For example, start with how the authors would recommend that people in other countries determine the size of the area that should be included in developing an Action Plan? How should relevant stakeholders be determined? What are the challenges in identifying who should be involved? How should appropriate assessments be developed so that quantitative data could be collected before and after implementation? How should workshops be designed to encourage stakeholder participation? Highlight what this pilot did well and where improvements could be made for future adoptees of the approach; while as the authors note this isn't a cost study, it might be helpful to indicate what would be ideal vs what would be necessary (or at least strongly recommended).

Reviewer #2: The first paragraph of the Discussion summarises the methodology into 6 steps. The first of these (GIS + community consultation to find all the settlements in the study area) is obvious, is reported well and is appropriately discussed. The next 2 steps are essentially quantitative in nature (counting people by census and by consulting treatment registers) but no quantitative data or analyses are presented. Without quantitative reporting, the interpretation and discussion of these steps is not credible. There is no evidence presented to support the conclusions that are drawn.

I was completely lost when I reached the point in the Discussion that reads “Better MDA quality and performance was achieved as a result of action plan implemented. A 25.2% (1,023 treated individuals) increase in the number of persons treated was achieved during the MDAs in July compared to January 2024”. I simply could not find these critical data that apparently report the coverage in July 2024 (noting again that the coverage data in the text are referred to as January 2024 and also as March 2024, but certainly not July). Where are they? None of the Discussion that follows concerning the success of this approach to improving MDA coverage has any meaning unless there is rigorous quantitative data to support the claimed increase in MDA coverage, and data (preferably quantitative structured survey data rather than anecdotal focal group discussion reports) that linked the observed increase in coverage to specific actions.

**Editorial and Data Presentation Modifications?**

Reviewer #1: Minor comments:

Background:

p4

remove parenthesis "onchocerciasis transmission by the year 2030)"

indicate when OCP ended

p5

6.7% and 5.9%

"2021 investigation revealed that the NTD" add "programme's"

"inability to reach some at-risk populations"

"including (illegal) miners, were"

remove also and fix typos: "over the years were contributing to persistent on-going"

"We determined in our 2021..."

Methods:

p7 Figure legend: "10 km radius delimits"

"treatment coverage evaluation surveys"

p10 "a month following preceding MDA" --wording is confusing. Is this a typo? A month after MDA but before implementation of the action plan? Both before and after MDA?

p11 "During the workshop, an action plan"

"Step 7: Implementation and evaluation of action plan" (not plans)

p12 "Commcare platform and cleaned and analysed using STATA"

Remove definition of percentage "Percentages were calculated as..."

(odds ratio, OR) not odd

Fix typo: "To ensure quality of transcription, research assistants that conducted"...

Results:

p12

add mdash "... Akura, and Mambeele)-were identified and included."

Already indicated settlements are remote; "These hamlets have poor accessibility..."

The text reads that these settlements are "often assimilated into bigger communities for CDTI". Because this statement refers to actual settlements, rather than small remote settlements in general, there should be data on how many of the ten settlements and in how many rounds of CDTI they were assimilated--i.e., quantitative rather than qualitative.

p 13

"The outcome of the first few days" -- first few days of what? I'm confused

Regarding the "shacks", I'm confused by this term--do people not live in shacks? I feel like this term is being used as if it is referring to a specific type of dwelling that is not a temporary shelter?

p14

"those who had died or moved away were still"

Capitalize Table 1

"uptake 0.26" is this a % or proportion? Looks like a proportion--I would recommend being consistent between reporting % or proportion in table 1 and text please.

p15

capitalize Table 2

Table 2--Coverage column: indicate what number is in (). Is this the number of people that were covered or the total number of people in the category? Does Age have a reference? Why is there no OR for each and only for age 35-44? Residency -- remove carriage return and uncapitalize "travel outside the district". Fix typos: *15 years cut-off age ensured a fair comparison across variables

p17

"assessment revealed context-specific challenges"

"interviewees which may have an influence"

p18

"community members included the belief that the disease is gotten from a contaminated"

Has "IDI" been defined?

The suggestion that cattle herders rarely participate in MDAs because of seasonal migration is not supported by the provided statement. Is this speculation by the authors or is it something that was in the interviews?

p19

"Community members reported that the absence"

Indent last quote on page for consistency

p20

Fix quote and spelling in second quote

Remove comma "participation in MDA in the communities is the fear"

"after taking the medicine and decided they would not take"

p21 - community members wanted more CDDs; how many is more? indicate how this was changed in the Action Plan if it was

"Participants suggested that the community should be informed"

Add quotes around last quote on page. Is this community Kwadum Nsuta, Kwadum Joe, or a third community?

p22

Is this a boy? can CDDs be under 18? or is this to indicate male?

"Community members therefore recommended" ..." This would ensure that the true"..."persons who do not take or refuse to swallow the drug would be easily identified"

"mobilized the community effectively and changed the community's"

The last paragraph of this page needs to be better organized. Start with what people were involved, then what the action plan included, then what was in the action plan.

"hard-to-reach groups, including"

consistent spelling for sensitization/sensitisation. I also don't know what the authors mean by this word no matter the spelling?

See detailed Action Plan in Appendix 3.

p 23

Implementation and evaluation of action plan -- this section is very thin, only one sentence! Perhaps remove subheading of qualitative assessment?

Census using electronic registration should be noted in the section above

Is there a bias in who comes to the FGD? When were the FGDs run relative to the implementation? What is the time scale of any of this?

capitalize Kwandum

p24

"Participants also highlighted areas for"

what is a "mop-up"? this is a fairly casual term where I come from. Perhaps "follow-up"?

check spelling of "wifse"

check capitalization of Kwadum

Add consistency with whether Action Plan is capitalized or not

p25

"no study has combined"

"identification of 12 settlements that were not identified in"

"masked the need for CDDs as well as straining the ability"

Be consistent in whether use blackflies/blackfly vs black flies/black fly [I prefer the former]

I'm not really sure what the term "reinforced treatment" means, or "assimilate to" or "sensitisation of the people". CDDs has already been defined previously.

2nd paragraph conflates identification of additional communities with the challenges of reaching people in settlements; please clean this up

p27

first paragraph -- I don't think the authors mean "quality of treatment"? Poor coverage perhaps? "Better quality has been achieved in terms of" -- not very well worded; "Coverage estimates have been improved because of..."? Or just "the Action Plan improved accuracy in drug..."? "This improvement will be impactful..." also needs some copyediting of this sentence please

Figures:

Figure 1-- Recommend adding an inset map of Ghana indicating the relevant area under consideration for context. The legend is very confusing. Why is parasitology a pie chart? Is this the proportion of people who were skin snip mf+ve? Why is parasitology a pie chart while serology is a colored range? Is the larvae found supposed to indicate mf or iL3 larvae, perhaps blackfly larvae?

Figure 2-- Typos in "analyse data" set. "gaps in MDA coverage, map areas with persistent infection, map areas with inadequate...". Shouldn't identifying barriers, solutions, and opportunities to participate be a separate step from data analysis, given that this is interpreting the data? Isn't this part meant to be done in consultation with key community people? Title: Develop Action-- add "Plan". Note that "enhance community engagement" assumes that this is a problem a priori, while it might not be in all areas and thus in some areas would be identified as a solution in the Action Plan. Typo "ensure continued community engagement"

Figure 3--legend indicates that this is a "transmission focus", but need to indicate in text the data that suggest that this 10km radius is a transmission focus and not an arbitrary delimitation. Copyright notice needs to be part of sentence that precedes it so that it doesn't mis-label the figure itself as (c) Esri.

Reviewer #2: (No Response)

**Summary and General Comments**

Reviewer #1: Onchocerciasis and other parasitic NTDs are close to elimination. However, hotspots of persistent transmission remain despite decades of drug distribution, and new solutions are needed. Nditanchou et al. have proposed an approach to addressing some of the problems that may be contributing to lack of successful MDA at the community level. This approach involves connecting community, health workers, and programme staff to identify what some of the problems are and to co-design a solution (an "action plan") that has clear accountability. Their argument is that such an approach will lead to improved MDA coverage.

I enjoyed the manuscript and would like to see the study published. However, because the study takes place in a small area in Ghana, the study itself is not what makes this paper interesting. I recommend that the authors strengthen their framing of this in the Introduction and Discussion more explicitly as an approach to solve a problem, using Kwanware-Ottou as a test for broader roll-out. The way it is currently written, the focus is more on the results from Kwanware-Ottou.

I further think that there needs to be more clarity in the Discussion with regards to the limitations are in their approach, and how they would recommend other programmes proceed in implementing this type of approach given their experiences (e.g., geographic scale, metrics for evaluating success, time frame). This would make the paper far more applicable to the broader NTD audience that the journal targets.

There is significant value in this study--for example, the description of how activities were designed to "develop capacity at every level" was great! Appendix 3 represented a clear summary and was very interesting. The amount of coordination and work that went into the project is to be commended.

Reviewer #2: I do not think this paper can be published in its current form. Leaving aside what I think are some serious inconsistnencies in the data that are reported, there is simply insufficient evidence presented to support most of the claims and conclusions beyond those that relate to better defining the location and size of the target population. Nothing can be concluded about the likely causes of the apparently low historic coverage (although the coverage information presented is of doubtful validity) and there are no data presented to support either the action plan described nor the claim that coverage improved as a result. Substantial revision is required if this paper is to be published.

To be more positive, I do agree that it is extremely important to investigate the causes for persistent hotspots of transmission. It is clear that such hotspots are emerging in many onchocerciasis endemic settings throughout Africa, particularly where annual MDA is the sole tool employed. Low therapeutic coverage may be a cause in some of these, but (a) a much more rigorously quantiative approach than that taken here is required to demonstrate conclusively that this is indeed the likely cause of persistent transmission and (b) more rigorous analysis of the outcome of remedial actions is likewise required that the entirely qualitative claim of success that is made in this paper. My strong recommendation to the authors is to move away from anecdotal, qualtiative, subjective reporting to the use of tools such as Nviva to summarise and analyse qualitative data in a more objective, quantitative fashion.

PLOS authors have the option to publish the peer review history of their article (what does this mean? ). If published, this will include your full peer review and any attached files.

**Do you want your identity to be public for this peer review?** For information about this choice, including consent withdrawal, please see our Privacy Policy .

Reviewer #1: No

Reviewer #2: No

**Figure resubmission:**

**Reproducibility:**



---

## [Editor Report · Decision Letter 1]

20 Jan 2026

Dear NDITANCHOU,

We are pleased to inform you that your manuscript 'Reaching the Last Mile with Ivermectin Mass Drug Administration against Onchocerciasis: The case of Kwanware-Ottou persistent transmission focus in the Wenchi Health District of Ghana' has been provisionally accepted for publication in PLOS Neglected Tropical Diseases.

Best regards,

Vito Colella, DVM, PhD

Academic Editor

Eva Clark

Section Editor

Shaden Kamhawi

co-Editor-in-Chief

Paul Brindley

co-Editor-in-Chief

---

## [Editor Report · Acceptance letter]

Dear NDITANCHOU,

We are delighted to inform you that your manuscript, "Reaching the Last Mile with Ivermectin Mass Drug Administration against Onchocerciasis: The case of Kwanware-Ottou persistent transmission focus in the Wenchi Health District of Ghana," has been formally accepted for publication in PLOS Neglected Tropical Diseases.

Best regards,

Shaden Kamhawi

co-Editor-in-Chief

Paul Brindley

co-Editor-in-Chief
